# Interleukin-1ß Attenuates Expression of Augmenter of Liver Regeneration (ALR) by Regulating HNF4α Independent of c-Jun

**DOI:** 10.3390/ijms24098107

**Published:** 2023-04-30

**Authors:** Jonas Nimphy, Sara Ibrahim, Rania Dayoub, Marion Kubitza, Michael Melter, Thomas S. Weiss

**Affiliations:** 1Children’s University Hospital (KUNO), University Hospital Regensburg, 93053 Regensburg, Germany; 2Center for Liver Cell Research, University Hospital Regensburg, 93053 Regensburg, Germany

**Keywords:** augmenter of liver regeneration, IL-1ß, inflammation, NAFLD, cholestasis, cytokine

## Abstract

Inflammasomes and innate immune cells have been shown to contribute to liver injury, thereby activating Kupffer cells, which release several cytokines, including IL-6, IL-1ß, and TNFα. Augmenter of liver regeneration (ALR) is a hepatotropic co-mitogen that was found to have anti-oxidative and anti-apoptotic properties and to attenuate experimental non-alcoholic fatty liver disease (NAFLD) and cholestasis. Additionally, hepatic ALR expression is diminished in patients with NAFLD or cholestasis, but less is known about the mechanisms of its regulation under these conditions. Therefore, we aimed to investigate the role of IL-1ß in ALR expression and to elucidate the molecular mechanism of this regulation in vitro. We found that ALR promoter activity and mRNA and protein expression were reduced upon treatment with IL-1ß. Early growth response protein-1 (Egr-1), an ALR inducer, was induced by IL-1ß but could not activate ALR expression, which may be attributed to reduced Egr-1 binding to the ALR promoter. The expression and nuclear localization of hepatocyte nuclear factor 4 α (HNF4α), another ALR-inducing transcription factor, was reduced by IL-1ß. Interestingly, c-Jun, a potential regulator of ALR and HNF4α, showed increased nuclear phosphorylation levels upon IL-1ß treatment but did not change the expression of ALR or HNF4α. In conclusion, this study offers evidence regarding the regulation of anti-apoptotic and anti-oxidative ALR by IL-1ß through reduced Egr-1 promoter binding and diminished HNF4α expression independent of c-Jun activation. Low ALR tissue levels in NAFLD and cholestatic liver injury may be caused by IL-1ß and contribute to disease progression.

## 1. Introduction

The liver plays a critical role in numerous physiological processes, including synthesis of proteins, hormones, and bile acids; catabolism of xenobiotics; and metabolism of carbohydrates, cholesterol, and lipids. Hepatocytes are the liver cell type that fulfil the majority of these functions, thereby reflecting their important role in maintaining metabolic homeostasis [1]. Because of this indispensable role in the body, the liver has a tremendous capacity for regeneration, including anti-oxidative and anti-apoptotic mechanisms [2,3]. During inflammation, in response to detrimental stimuli, such as tissue injury or infection, immune cells are recruited to the site of injury [4]. This inflammatory response is essential for tissue repair, control of infection, and adaptation to different kinds of stress. Imbalance or disturbance of inflammatory reactions can play a major role in disease development, such as in hepatic fibrogenesis [5].

Liver diseases, such as non-alcoholic steatohepatitis (NASH) [6] and cholestasis [7], are often accompanied by inflammation, with elevated IL-1ß serum levels. NASH is mainly caused by the accumulation of free fatty acids in hepatocytes, resulting in lipo-toxicity and apoptosis. Pathogen-associated molecular patterns (PAMPs) and damage-associated molecular patterns (DAMPs), which are released by dead or damaged cells (e.g., ATP and uric acid), induce inflammasomes in Kupffer cells [4,8]. Activated Kupffer cells express IL-1ß as an inactive precursor, which is subsequently cleaved by caspase-1 (activated by inflammasomes) into active IL-1ß, a mediator and effector of inflammation [4,8]. Cholestatic liver diseases occur when the excretion of bile acids is impaired due to either direct inhibition of bile flow or genetic defects impairing bile acid transporters [9]. Whatever the cause, hepatic bile acid accumulation causes liver injury by inducing apoptosis [10] and manipulating mitochondrial membrane stability through oxidative stress and reactive oxygen species (ROS) production [11], as well as attracting inflammatory cells [12]. Similar to NASH, cholestasis may lead to liver fibrosis, cirrhosis, and possibly liver failure if left untreated [13]. Part of the explanation for the raised serum concentration of IL-1ß in this disease is the activation of Kupffer cells by IL-17 (released by T helper cells and neutrophils during cholestasis) [14,15], which leads to the induction of TNFα and IL-1ß [16]. Therefore, these findings indicate that IL-1ß, mainly released by Kupffer cells, may contribute to liver injury in NASH and cholestasis.

Augmenter of liver regeneration (ALR), a member of the ALR/Erv1 protein family, has been shown to augment the process of liver regeneration [17]. ALR is constitutively expressed in hepatocytes and cholangiocytes [18] and was previously shown to attenuate oxidative stress [19,20], free fatty acid-induced lipo-apoptosis [21,22], and endoplasmic reticulum (ER) stress [21,22]. Furthermore, deletion of ALR expression in mice leads to steatohepatitis and hepatocellular carcinoma [23,24]. Several reports have underlined the important role of ALR as an anti-apoptotic, anti-oxidative, and cell survival factor after injury (reviewed in [25,26,27]). Interestingly, although reduced hepatic ALR expression levels were found in liver diseases, such as cholestasis [28] and NASH [21], the underlying molecular mechanism for its regulation under these conditions is less clear. Cholestasis and NASH are both associated with elevated IL-1ß expression. However, the regulation of ALR expression by IL-1ß has not yet been investigated. Therefore, in this study, ALR mRNA and protein expression upon IL-1ß treatment in vitro were analyzed, and the involved transcription factors were evaluated.

## 2. Results

### 2.1. IL-1ß Reduces ALR Promoter Activity and Expression

To analyze the activity of the ALR promoter upon IL-1ß treatment, HepG2 cells were transfected with a full-length ALR promoter construct (−733/+527) [29] and subsequently treated with IL-1ß for different time points (Figure 1A). ALR promoter activity was significantly reduced after IL-1ß treatment. Furthermore, mRNA expression of ALR was reduced upon treatment with IL-1ß in hepatoma cells, HepG2 cells, Huh7 cells, and primary human hepatocytes (PHH) (Figure 1B). This was further confirmed for ALR protein expression by performing Western blotting. ALR expression was reduced after treatment with 10 or 25 ng/mL IL-1ß (Figure 1C), suggesting that IL-1ß diminishes the expression of ALR.

### 2.2. IL-1ß Decreases ALR While Alleviating HNF4α Expression

The *growth factor erv1-like* (GFER) gene (coding for ALR) promoter encompasses binding sites for several transcription factors, including HNF4α [25], which has been shown to induce ALR expression [30]. Therefore, mRNA expression of HNF4α in hepatoma cells and PHHs, treated with IL-1ß, was analyzed, and it revealed a significant reduction in HNF4α mRNA expression in Huh7 cells and PHHs (Figure 2A). Western blot analyses performed using lysates from hepatoma cells treated with IL-1ß confirmed reduced HNF4α expression upon IL-1ß treatment (Figure 2B). Reduced HNF4α protein levels were observed in total protein lysates, as well as in nuclear fractions, upon IL-1ß treatment. Decreased ALR expression upon IL-1ß treatment (Figure 1) was associated with lower HNF4α expression (Figure 2C). Interestingly, HNF4α over-expression, caused by transient transfection with an HNF4α expression plasmid, increased ALR expression; this increase could not be subsequently reversed by IL-1ß treatment (Figure 2C). These data suggest that IL-1ß decreases ALR expression, at least partially, due to reduced HNF4α expression.

### 2.3. IL-1ß Reduced Egr-1-Mediated ALR Induction

Egr-1, a zinc finger transcription factor, induces ALR expression by binding to its response element within the ALR promoter [30]. IL-1ß has been shown to increase the expression of Egr-1 [31,32]. In this study, we analyzed Egr-1 expression in HepG2 and Huh7 cells and found significantly enhanced Egr-1 mRNA expression (Figure 3A) and induced Egr-1 protein expression after 2–4 h upon IL-1ß treatment (Figure 3B). Therefore, the effect of Egr-1 on ALR promoter activity upon IL-1ß treatment in HepG2 cells was analyzed. The ALR promoter construct was co-transfected with an Egr-1 expression plasmid, and promoter activity was analyzed upon treatment with IL-1ß for different time points. As shown in Figure 3C, Egr-1 without IL-1ß treatment activated ALR promoter activity. This activity was then significantly reduced after IL-1ß application, starting 4 h after treatment (Figure 3C). This was further confirmed by performing EMSA to analyze Egr-1 binding to its response element within the ALR promoter after IL-1ß treatment. This binding was reduced after the application of IL-1ß (Figure 3E, lane 4). In addition, expression of specificity protein 1 (SP1), an inducer of ALR promoter activity [33], was reported to be decreased in chondrocytes upon IL-1ß treatment [31,34]. However, mutating the overlapping SP1 response element next to the Egr-1 response element [31,35] (Figure 3D) increased the DNA–protein binding of Egr-1 to the ALR promoter (Figure 3E, lane 5). This binding was nevertheless reduced upon IL-1ß treatment (Figure 3E, lane 6). In summary, Egr-1 binds to its binding site within the ALR promoter (+304/+314), and this binding is reduced by IL-1ß and therefore supposedly responsible for ALR suppression. However, this reduced binding of Egr-1 seems to be SP1 independent, since IL-1ß potentially decreases SP1 expression, and the mutating SP1 binding site increases Egr-1 binding within the ALR promoter.

### 2.4. Diminished ALR Expression Is Independent of c-Jun Induction by IL-1ß Treatment

It has been reported that c-Jun binds to the AP-1/AP-4 binding site within the ALR promoter and consequently represses ALR promoter activity [36]. Therefore, we treated HepG2 and Huh7 cells with IL-1ß and found enhanced c-Jun (a key member of the AP-1 family) mRNA expression, but not AP-4 mRNA expression, in hepatoma cells (Figure 4A). Nuclear localization and activation (by phosphorylation) of c-Jun by IL-1ß was confirmed using Western blot, showing distinct bands of phosphorylated c-Jun upon IL-1ß treatment in HepG2 and Huh7 cells in nuclear extracts (Figure 4B). Furthermore, silencing and enhancing c-Jun expression by transfection experiments with c-Jun siRNA or c-Jun expression plasmids revealed unaltered ALR mRNA or ALR protein expression without IL-1ß treatment (Figure 4C,D). It has previously been shown that IL-1ß decreases HNF4α mRNA and protein expression, while increasing phosphorylated c-Jun [37]. In our study, while silencing c-Jun expression, IL-1ß treatment did not reduce HNF4α mRNA levels, but HNF4α protein expression was diminished upon IL-1ß treatment independent of c-Jun silencing or over-expression (Figure 4C,D). ALR mRNA and protein expression were reduced by IL-1ß treatment irrespective of c-Jun silencing or over-expression (Figure 4C,D). In summary, reduced ALR expression caused by IL-1ß seems to be, at least in part, mediated by HNF4α independent of c-Jun.

## 3. Discussion

In this study, we present evidence for the negative regulation of ALR expression by IL-1ß. ALR mRNA and protein expression were reduced upon IL-1ß treatment, which is partially mediated by diminished HNF4α expression and HNF4α nuclear localization and also possibly by reduced binding of Egr-1 to the ALR promoter. On the contrary, the transcription factors c-Jun and AP-4 did not reduce ALR expression per se and did not seem to play a role in the ALR reduction caused by IL-1ß.

Augmenter of liver regeneration (ALR) is an anti-apoptotic and ER stress-reducing protein [21,22]. Furthermore, ALR is activated by nuclear factor erythroid 2-related factor 2 (Nrf2) binding to the anti-oxidative response element (ARE) within the ALR promoter upon exposure to oxidative stress [38] and is therefore considered anti-oxidative [19,20,38]. Oxidative stress and its resulting lipid peroxidation products may activate proinflammatory cytokines and are involved in fibrogenesis. Therefore, altering the anti-oxidative capacity may prevent the activation of pro-inflammatory cytokines [39]. Moreover, it has been shown that endogenous short-form ALR (15 kDa) increases signal transducer and activator of transcription 3 (STAT3) activation [40] and thereby alters IL-6 signaling in hepatocytes. Interestingly, it has previously been demonstrated that STAT3 activation had a hepatoprotective role in a severe NASH model [41] and that STAT3 is a negative regulator of bile acid synthesis and provides protection from bile acid-induced apoptosis [42]. Additionally, STAT3 activation reduced the expression of death receptor 5 (DR5, TRAIL-receptor) [43]. Therefore, enhancing the expression of ALR in hepatocytes may increase cellular anti-oxidative defense, lower the activation of pro-inflammatory cytokines, and might protect hepatocytes against apoptosis during NASH or cholestasis.

However, little is known about the regulation of ALR expression by Kupffer cell-released IL-1ß. We assessed the regulation of ALR by IL-1ß using luciferase assays, qRT-PCR, and Western blot techniques and found that ALR expression and promoter activity are reduced by IL-1ß. It has previously been reported that IL-1ß can induce AP-1 expression [44,45] and that c-Jun binds to an AP-1/AP-4 binding site (−375/−369) within the ALR promoter, repressing its activity in HepG2 cells, based on a promoter study [36]. In this study, we were able to show that IL-1ß induces c-Jun mRNA expression and c-Jun phosphorylation, as well as nuclear localization, in several hepatoma cells, in contrast to AP-4. Interestingly, in contrast to the results obtained from the referenced promoter study [36], silencing or enhancing c-Jun expression did not change ALR mRNA or protein expression, as well as its IL-1ß-mediated reduction. Therefore, ALR expression and its reduction by IL-1ß are independent of c-Jun expression. Of note, blunted c-Jun expression reduces cell proliferation, and c-Jun knockout mice show impaired liver regeneration, mainly based on down-regulation of tumor suppressors p53 and p21 and their transcriptional activities. [46,47] The findings of c-Jun-independent ALR expression in this study are not contradictory to these reports, unlike the results from the ALR promoter study [36], and underline the importance of the protein expression analysis that was performed.

Furthermore, it was shown that IL-1ß decreases HNF4α mRNA and protein expression while increasing phosphorylated c-Jun levels in human liver cells [37,48]. Additionally, we have reported previously that HNF4α1, but not HNF4α7, can induce ALR expression [30]. We confirmed our finding by showing increased ALR expression upon HNF4α over-expression. On the other hand, IL-1ß treatment decreased HNF4α expression in the presence or absence of phosphorylated c-Jun. Therefore, we found in this study that HNF4α expression and its regulation by IL-1ß are independent of c-Jun phosphorylation. Thus, the reduction in ALR expression by IL-1ß is mediated by HNF4α independent of c-Jun activation.

Expression of Egr-1, an inducer of ALR expression [30], is enhanced by IL-1ß [31,32], which was confirmed in our study of Egr-1 mRNA and protein expression. Induction of Egr-1 expression by IL-1ß involves activation of the EGF receptor by metalloproteinase (MMP) and ADAM (a disintegrin and a metalloproteinase), leading to activation of the MAPK/ERK1/2 pathway and, consequently, to Egr-1 transcription. In addition, it has been shown that c-Jun regulates Egr-1 [45] and that the loss of HNF4α (HNF4α knockout mouse model) induces Egr-1 expression [49]. We have previously shown that Egr-1 induces ALR expression [30] and shown in this study that Egr-1 expression increases ALR promoter activity in the absence of IL-1ß but decreases ALR promoter activity in the presence of IL-1ß. This is further underlined by EMSA analysis, demonstrating reduced Egr-1 binding to the ALR promoter upon IL-1ß treatment. While IL-1ß activates c-Jun [45], reduces HNF4α expression [49], and therefore leads to increased Egr-1 expression in the presence of IL-1ß, binding of Egr-1 to the ALR promoter and possibly ALR expression are reduced. Notably, it has been shown before that IL-1ß is able to increase binding of Egr-1 to respective binding sites on PPARγ [31] and type II collagen (COL2A1) [50] promoters in chondrocytes. In summary, IL-1ß enhances Egr-1 expression but reduces ALR expression, at least in part, by reducing Egr-1 binding to the ALR promoter via an unknown mechanism.

To shed more light on the role of ALR in liver disease development, studies have been performed in which ALR was silenced and the outcome of its absence was monitored [25]. Briefly, liver-specific ALR knockout mice develop steatosis, accompanied by enhanced ROS production and inflammatory cell recruitment to the liver [23]. Furthermore, ALR deficiency has been shown to be a critical predisposing factor for NASH and fibrogenesis [51]. These findings are supported by reports showing that enhanced ALR expression reduces the severity of fatty acid-associated injury [52] and diminishes ER stress, as well as lipo-apoptosis [21,22]. Additionally, patients with NASH have reduced hepatic ALR expression [21,51] and lower serum ALR levels [51]. Therefore, ALR plays an important role in NASH/fibrosis development, but the mechanism regulating ALR expression under these conditions is less clear. We and others found reduced expression of FOXA2 (HNF3ß), a transcription factor that positively regulates the expression of ALR [29], in the hepatic tissues of NASH patients [21,53]. In addition, here we present evidence that ALR is negatively regulated by IL-1ß, a cytokine that is known to be activated by inflammasomes, as well as oxidative stress, and is involved in the pathogenesis of NASH [8]. In summary, it is likely that diminished ALR expression in NASH could be regulated by IL-1ß and FOXA2, which may accelerate disease progression.

Inflammation and oxidative stress are also observed in another common liver disease, cholestasis. Cholestasis is a pathological syndrome caused by impaired or disrupted bile flow from the liver, resulting in hepatic apoptosis [10], oxidative stress [11], and the attraction of inflammatory cells [12]. Furthermore, IL-17, released from T helper cells, activates liver-resident macrophages (Kupffer cells) [14,15], which in turn release several pro-inflammatory cytokines, including IL-1ß, IL-6, and TNFα [16]. Furthermore, IL-1ß receptor knockout mice have demonstrated insusceptibility against LPS challenge after BDL (bile duct ligation), secreting fewer cytokines and showing lower mortality compared with wild-type mice [54]. These studies indicate that Kupffer cells and their related cytokines are associated with mortality in cholestatic liver injury [55]. A recent report identified IL-1ß signaling and, consequently, NFκB activation as a requirement for parenteral nutrition-associated cholestasis (PNAC) [56]. The authors showed that the reduction in BSEP and MRP2 expression in their mouse model by IL-1ß may be causative for PNAC [56]. PNAC is associated with enhanced IL-1ß levels in pediatric patients [57] but, in addition, cholestatic patients with PBC or PSC showed increased IL-1ß and activated inflammasomes [58]. We hypothesize that diminished ALR expression in patients with cholestasis [28] may depend, at least in part, on elevated IL-1ß levels.

Inflammation in NASH and cholestasis is characterized not only by enhanced levels of IL-1ß but also by increased release of IL-6 and TNFα [16,55]. IL-6 was shown to enhance ALR expression by increasing the binding of FOXA2 to the ALR promoter [29]. On the other hand, FOXA2 expression and activity are low in NASH [21,53] and, therefore, high IL-6 levels might be without consequences. Additionally, TNFα was demonstrated to increase the release of ALR in an in vitro model [59] and therefore might contribute to low cellular levels in NASH and cholestasis.

While most researchers have focused on the role of ALR in disease development by performing loss and gain of function models, less is known about the molecular mechanisms of ALR regulation. Here, we present evidence that IL-1ß reduces ALR mRNA and protein expression through diminished HNF4α expression and decreased Egr-1 binding to the ALR promoter independent of c-Jun activation. ALR was reported to have anti-apoptotic and anti-oxidative properties, resulting in less oxidative stress and presumably less induction of pro-inflammatory cytokines. Even though ALR expression is induced by oxidative stress, excessive levels of IL-1ß might impede proper ALR expression for cell protection and regeneration. Low ALR expression levels have been reported in liver tissue from patients with NASH and cholestasis, and the observed high IL-1ß levels in these patients may be causative. Further studies investigating the impact of IL-1ß on ALR’s hepatoprotective capacity, expression of ALR under inflammatory conditions, and correlations between IL-1ß and ALR levels in patients would be of interest.

## 4. Materials and Methods

*Cell culture and treatments*: Cells from the human hepatoma cell line HepG2, obtained from the American Type Culture Collection (HB-8065, ATCC, Manassas, VA, USA), and Huh-7 cells (ECACC 01042712), obtained from the European Collection of Authenticated Cell Cultures (ECACC) (Salisbury, UK), were grown at 37 °C, 5% CO_2_ in DMEM (BioWhittaker, Verviers, Belgium) supplemented with penicillin (100 units/mL), streptomycin (10 μg/mL), and 10% fetal calf serum (Biochrom, Berlin, Germany). Cells were seeded at a density of 4 × 10^4^ cells/cm^2^ and cultured for 24 h, followed by 18–24 h starvation (FCS-free conditions) as indicated and treated with the indicated concentrations of IL-1ß (PeproTech GmbH, Hamburg, Germany). Primary human hepatocytes (PHH) were isolated and maintained in culture, as described earlier [28,60], and analyzed using qRT-PCR.

*Reporter gene assays:* HepG2 cells were cultured at a density of 5 × 10^4^ cells/cm^2^ and then transfected using Lipofectamine^®^ 3000 (ThermoFisher Scientific, Darmstadt, Germany) with 200 ng of human ALR promoter construct (−733/+527) [29]. Additionally, 200 ng of either pcDNA3.1 or pcDNA3.1-Egr-1 expression plasmids (a generous gift from Xiaojia Chen, China) was transfected. After 24 h, IL-1ß was added for the indicated times. Dual luciferase assays were carried out 24 h after transfection using a dual luciferase reporter assay system (Promega, Mannheim, Germany). The pRL-TK Renilla vector was co-transfected to determine transfection efficiency, and the promoterless vector pGL_2_-basic served as a negative control. Each experiment was repeated at least three times.

*Transient transfection with HNF4α and c-Jun and silencing c-Jun expression:* HepG2 cells were transfected using Lipofectamine^®^ 3000 (ThermoFisher Scientific, Darmstadt, Germany) with 2.5 µg of pcDNA3.1-HNF4α (received thankfully from Dr. Oliver Burk, Stuttgart), pcDNA3.1-c-Jun-V5 (a generous gift from R.A. Cerione, Cornell University, USA) [61], or 30 pmol c-Jun si-RNA (# sc-29223; Santa Cruz, CA, USA). Transfected cells were maintained in culture for 24 h and then treated with 10 ng/mL IL-1ß for 24 h. The cells were lysed, and mRNA and/or total protein were isolated for further analyses.

*RNA isolation, reverse transcription, and qRT-PCR:* Total RNA was isolated using an RNeasy Mini Kit (Qiagen, Hilden, Germany). One microgram of total RNA was reverse-transcribed using a Reverse Transcription System from Qiagen (Hilden, Germany). Transcription levels of ALR, HNF4α, Egr-1, SP1, c-Jun, AP-4, and HPRT were quantified using real-time PCR technology (Roche, Penzberg, Germany). Primer pairs were designed as reported earlier (see Table 1) or with Primer3Plus (www.primer3plus.com, accessed on 24 January 2016). PCR reaction products were verified using sequence analysis, and each quantitative PCR was performed in triplicate. The primers used for qRT-PCR are listed in Table 1.

*SDS-PAGE and immunoblotting:* Total proteins were isolated, prepared, and subjected to Western blot analysis. Briefly, 30 μg per lane was separated by 12% SDS-PAGE (Biorad, Hercules, CA, USA), and proteins were transferred onto PVDF membranes (Biorad, Hercules, CA, USA), incubated with specific antibodies, and developed using an enhanced chemiluminescence reagent (ThermoFisher Scientific, Darmstadt, Germany). Anti-ALR polyclonal antibody was “custom made” against 15 kDa ALR (short form ALR) prepared by Davids Biotechnology (Regensburg, Germany). The following antibodies were used: anti-ß-actin (#4970); anti-HNF4α (#3113) obtained from Cell Signaling (Danvers, MA, USA); anti-HDAC (#05–100) purchased from Merck (Darmstadt, Germany); and anti-Egr1 (#189), and anti-phospho-c-Jun (#822) purchased from Santa Cruz Biotechnology (Heidelberg, Germany). Secondary goat HPR-conjugated antibodies (anti-rabbit #P0448 and anti-mouse #P0447) were obtained from Dako (Hamburg, Germany).

*Electrophoretic mobility shift assay (EMSA):* EMSA was performed as published earlier [30]. Briefly, cells were transfected with 500 ng of Egr-1 expression plasmid, and nuclear extracts were prepared using NE-PER™ Nuclear and Cytoplasmic Extraction Reagents (ThermoFisher Scientific, Darmstadt, Germany), according to the manufacturer’s description. Complementary synthetic biotin-labeled oligonucleotides corresponding to the Egr-1 binding site (+304/+314) within the ALR promoter were obtained from Metabion (Martinsried, Germany): Egr-1-RE Fwd: 5′-cctgtccccgcccccgcccaggta-3′ and Egr-1-RE Rev: 5′-tacctgggcgggggcggggacagg-3′. For the mutated overlapping SP1 binding site, the following oligonucleotides were used (mutated nucleotides are underlined): Egr-1-ΔSP1-Fwd: 5′-cctgtccttgcccccgcccaggta-3′ and Egr-1-ΔSP1-Rev: 5′-tacctgggcgggggcaaggacagg-3′. The biotin-labeled DNA was then detected using an enhanced chemiluminescence reagent (ThermoFisher Scientific, Darmstadt, Germany). Competition experiments were performed using a 200-fold excess of the same unlabeled binding site 20 min prior to electrophoresis using 6% DNA retardation gels (Biorad, Hercules, CA, USA) at room temperature.

*Statistical analysis:* All data are presented as mean +/− standard deviation or standard error of the mean as indicated. Data were compared between groups using Student’s *t*-test or ANOVA with post-hoc Bonferroni correction where appropriate. Values of *p* < 0.05 were considered significant (SPSS Statistics 25.0 program, IBM, Leibniz Rechenzentrum, München, Germany).

## Figures and Tables

**Figure 1 ijms-24-08107-f001:**
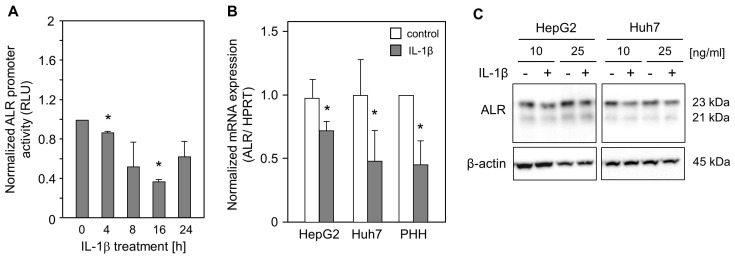
IL-1ß reduces ALR promoter activity and ALR expression. (**A**) HepG2 cells were transfected with ALR promoter construct (−733 to +527 bp), starved for 24 h, and then treated with 50 ng/mL IL-1ß for the indicated time points. Cells were then lysed to measure luciferase expression (three independent experiments, mean ± SD). (**B**) HepG2 and Huh7 cells were treated without (control) or with IL-1ß (10 ng/mL) for 24 h (five independent experiments). Primary human hepatocytes (PHHs) were starved for 24 h, followed by treatment without (control) or with IL-1ß (2 ng/mL) for 8 h (three independent experiments). The mRNA levels of ALR were analyzed and normalized to HPRT (mean ± SD). * *p* < 0.05 was considered significant compared to non-treated cells. (**C**) Western blot analysis using specific anti-ALR antibodies demonstrated decreased ALR protein expression in HepG2 and Huh7 cells upon treatment with IL-1ß (10 or 25 ng/mL) for 24 h. ß-actin served as the loading control. One representative blot is shown.

**Figure 2 ijms-24-08107-f002:**
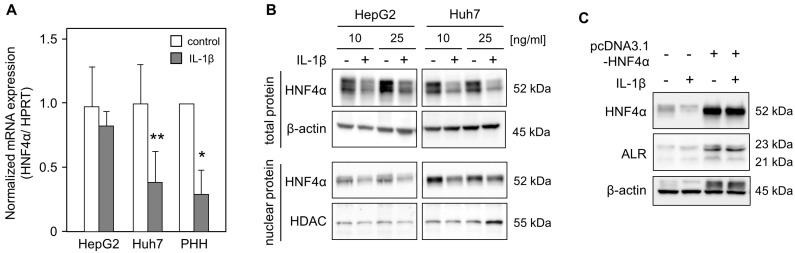
IL-1ß reduces HNF4α expression, while HNF4α induces ALR expression. (**A**) HepG2 and Huh7 cells were treated without (control) or with IL-1ß (10 ng/mL) for 24 h (five independent experiments). PHHs were starved for 24 h, followed by treatment without (control) or with IL-1ß (2 ng/mL) for 8 h (three independent experiments). The mRNA expression of HNF4α was analyzed using qRT-PCR and normalized to HPRT (mean ± SD). ** *p* < 0.01 or * *p* < 0.05 were considered significant compared to non-treated cells. (**B**) HepG2 and Huh7 cells were incubated with IL-1ß (10 or 25 ng/mL) for 24 h following the isolation of total and nuclear protein extracts. Western blot analysis using specific anti-HNF4α antibodies was performed with ß-actin or HDAC as loading controls. One representative blot of each cell line is shown. (**C**) HepG2 cells were transfected for 24 h with an HNF4α expression plasmid and then treated with IL-1ß (10 ng/mL) for 24 h. Cells were lysed to analyze protein expression using Western blotting. One representative blot is illustrated, with ß-actin serving as the loading control.

**Figure 3 ijms-24-08107-f003:**
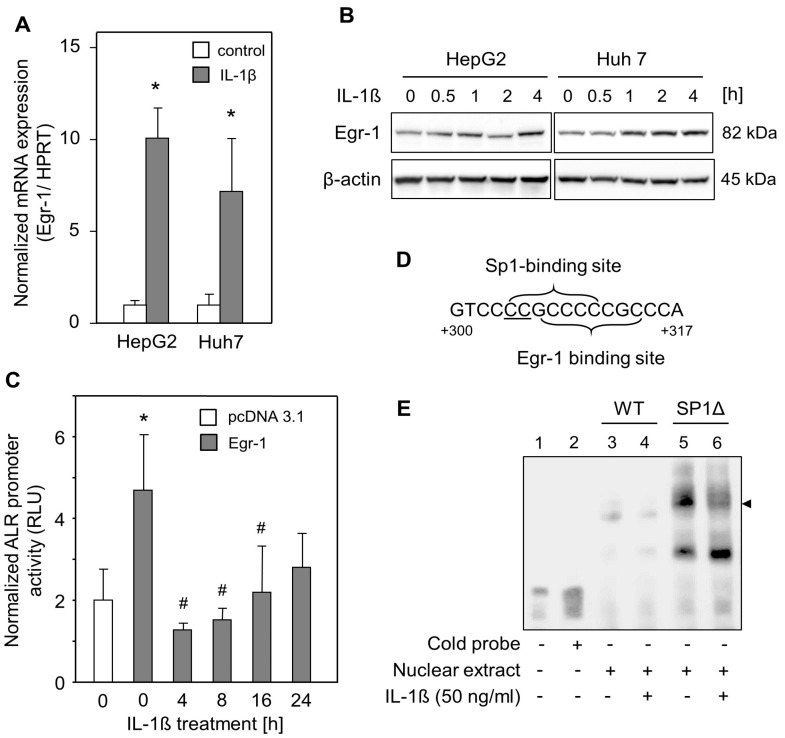
IL-1ß attenuates Egr-1 binding to the ALR promoter independent of SP1. (**A**) HepG2 and Huh7 cells were incubated without (control) or with IL-1ß (10 ng/mL) for 2 h, and Egr-1 mRNA expression was analyzed with qRT-PCR followed by normalization to HPRT (three independent experiments each, mean ± SD). * *p* < 0.05 was considered significant compared to non-treated cells. (**B**) HepG2 and Huh7 cells were incubated with IL-1ß (10 ng/mL) for the indicated times. Analysis by Western blotting showed increased Egr-1 expression from 1 to 4 h after IL-1ß incubation. ß-actin served as the loading control. (**C**) ALR promoter activity was measured after transfection of HepG2 cells with Egr-1 expression plasmids following starvation for 24 h and then the addition of IL-1ß (50 ng/mL) for the indicated time points (three independent experiments, mean ± SD). * *p* < 0.05 was considered significant compared to empty vector transfected cells; # *p* < 0.05 was considered significant compared to Egr-1 transfected and non-treated cells. (**D**) Schematic illustration of the overlapping binding sites of Egr-1 and SP1, a known repressor of ALR expression, within the ALR promoter (+300/+317). Both mutations (C→T) generated in the SP1 binding site are underlined. (**E**) Labeled oligonucleotides (+304/+314) representing Egr-1 and overlapping wild-type SP1 binding site (WT) or mutated SP1 binding site (SP1Δ) were used for EMSA. Oligonucleotides were incubated in the absence or presence of nuclear extracts derived from HepG2 cells with or without IL-1ß (50 ng/mL) treatment for 15 min. For competition experiments, a 200-fold molar excess of unlabeled Egr-1 consensus oligonucleotides was added (lane 2). Egr-1 specifically binds to the Egr-1 binding site within the ALR promoter (lane 3). Treatment of cells with IL-1ß decreased the binding of Egr-1 to its response element (lane 4). Lanes 5 and 6 represent the ΔSP1 oligonucleotides, resulting in increased Egr-1 binding to DNA upon SP1 mutation (lane 5). This binding is nevertheless reduced after the addition of IL-1ß addition (lane 6).

**Figure 4 ijms-24-08107-f004:**
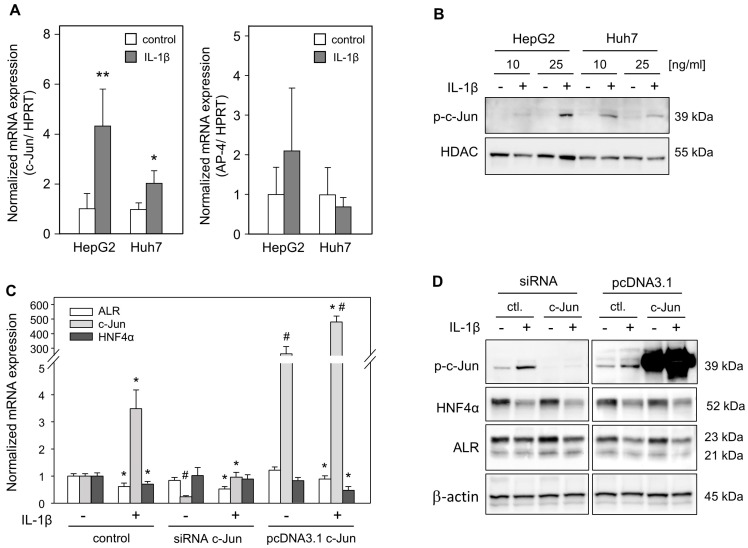
IL-1ß enhances c-Jun expression and nuclear localization, but c-Jun does not affect ALR expression. (**A**) HepG2 and Huh7 cells were incubated without (control) or with IL-1ß (10 ng/mL) for 24 h. The mRNA expression of c-Jun and AP-4 were analyzed using RT-PCR with normalization to HPRT (five independent experiments each, mean ± SD). * *p* < 0.05, ** *p* < 0.01 were considered significant compared to non-treated cells. (**B**) HepG2 and Huh7 cells were incubated with IL-1ß (10 or 25 ng/mL) for 24 h, and nuclear extracts were isolated. Western blot was performed using specific anti-phospho-c-Jun antibodies, with HDAC serving as the loading control. (**C**) Huh7 cells were transfected for 24 h with empty vectors (control), c-Jun siRNA, or c-Jun expression plasmids (pcDNA3.1 c-Jun) and treated without (0 ng/mL) or with IL-1ß (10 ng/mL) for 24 h. The mRNA expression of ALR, c-Jun, and HNF4α were analyzed using RT-PCR with normalization to HPRT (n = 3, mean ± SEM). * *p* < 0.05 was considered significant compared to non-IL-1ß-treated cells. # *p* < 0.05 was considered significant compared to the corresponding control. (**D**) Huh7 cells were treated as described in (**C**) and were lysed to analyze p-c-Jun, ALR, and HNF4α protein expression using Western blotting with specific antibodies. One representative blot is illustrated, with ß-actin serving as the loading control.

**Table 1 ijms-24-08107-t001:** Primer sequencenesused in qRT-PCR experiments.

ALR, sense	5′-gaagcgggacaccaagttta-3′	[28]
ALR, antisense	5′-ttcagcacactcctcacagg-3′	
SP1, sense	5′-ttgaaaaaggagttggtggc-3′	[62]
SP1, antisense	5′-tgctggttctgtaagttggg-3′	
Egr-1, sense	5′-tactcctctgttccccctgctt-3′	
Egr-1, antisense	5′-gaaaaggttgctgtcatgtccg-3′	
HNF4α, sense	5′-tgtcccgacagatcacctc-3′	[28]
HNF4α, antisense	5′-cactcaacgagaaccagcag-3′	
c-Jun, sense	5′-gcaaacctcagcaacttcaacc-3′	[63]
c-Jun, antisense	5′-gcatctcgggcactgtctga-3′	
AP-4, sense	5′-gtgcccactcagaaggtgc-3′	[64]
AP-4, antisense	5′-ggctacagagccctcctatca-3′	
HPRT, sense	5′-tgacactggcaaaacaatgca-3′	[28]
HPRT, antisense	5′-ggtccttttcaccagcaagct-3′	

## Data Availability

All relevant data are within the paper.

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
