# Peer review of "Interleukin-1ß Attenuates Expression of Augmenter of Liver Regeneration (ALR) by Regulating HNF4α Independent of c-Jun"

_ijms, 2023, doi:10.3390/ijms24098107_

Round 1

Reviewer 1 Report

I carefully read the manuscript ”Interleukin 1β attenuates expression of Augmenter of Liver Regeneration (ALR) by regulating HNF4α but independent of c-Jun” by Jonas Nimphy et al.

The data reported of new metabolic pathways/regulation of ALR would lead to clarify and justify both many of the different molecular activities of ALR and to clarify some of the physiological aspects of tissue inflammation and the evolution of the fibrotic process, physiological situations identified not only in liver tissue.. In addition, the present data not only could open new fields of study but could be able to induce new therapeutic designs proposing a role for a molecule with anti-inflammatory, anti-apoptotic and anti-oxidative activities.

Minor suggestions

I would like better considerations and discussion on Egr-1 and IL-1ß relationship too briefly discussed by the Authors in lines 237-240.

I consider it appropriate to carry out a more detailed analysis of the bibliographic references on ALR.

Author Response

Reviewer #1:

  • “I would like better considerations and discussion on Egr-1 and IL-1ß relationship too briefly discussed by the Authors in lines 237-240.”

We now have included this information in the edited version of our manuscript and re-wrote the paragraph in the discussion:

“Expression of Egr‑1 , an inducer of ALR expression [30], is enhanced by IL-1ß [31,32], which is confirmed in our study for Egr-1 mRNA and protein expression. Induction of Egr-1 expression by IL-1ß involves activation of EGF receptor by MMP (metalloproteinase) and ADAM (a disintegrin and a metalloproteinase) leading to activation of MAPK/ERK1/2 pathway and consequently to Egr-1 transcription [49]. In addition, it has been shown that c-Jun regulates Egr-1 [45] and that loss of HNF4α (HNF4α knock-out mouse model) induces Egr-1 expression [50]. We have shown earlier, that Egr-1 induces ALR expression [30] and show in this study, that Egr‑1 expression increases ALR promoter activity in absence of IL-1ß, but reduces ALR promoter activity in the presence of IL-1ß. This is further underlined by EMSA analysis demonstrating reduced Egr‑1 binding to ALR promoter upon IL-1ß treatment. While IL-1ß activates c-Jun [45], reduces HNF4α expression [50] and therefore leads to increased Egr-1 expression, in presence of IL-1ß binding of Egr-1 to ALR promoter and possibly ALR expression are reduced. Notably, it was shown before that IL-1ß is able to increase binding of Egr-1 to respective binding sites on PPARγ [31] and typ II collagen (COL2A1) [51] promoters in chondrocytes.“

  • “I consider it appropriate to carry out a more detailed analysis of the bibliographic references on ALR.”

We now have included this information in the edited version of our manuscript and re-wrote the paragraph in the introduction and updated some references in the discussion:

“Augmenter of Liver Regeneration (ALR), a member of the ALR/Erv1 protein family, was shown to augment the process of liver regeneration [17]. ALR is constitutively expressed in hepatocytes and cholangiocytes [18] and was previously shown to attenuate oxidative stress [19,20], free fatty acid-induced lipo-apoptosis [21,22] and ER (endoplasmatic reticulum)-stress [21,22]. Furthermore, deletion of ALR expression in mice leads to steatohepatitis and hepatocellular carcinoma [23,24]. Several reports underlined the important role of ALR as an anti‑apoptotic, anti-oxidative and cell survival factor after injury (reviewed in [25-27]). Interestingly, while reduced hepatic ALR expression levels were found in liver diseases like cholestasis [28] or NASH [21], the underlying molecular mechanism for its regulation under these conditions is less clear.”

Reviewer 2 Report

The work presented by the authors looks very interesting. However, it needs corrections to minor methodological errors and text editing. As a result, before publication, the manuscript requires minor revision. I want to review the manuscript's revised version before it is accepted for publication.

Comment 1- Start the introduction section with a brief role of the liver in metabolic homeostasis.

https://doi.org/10.1155/2021/6661937.

Comment 2- Oxidative stress plays an important role in liver cirrhosis and related pathologies. The lipid peroxidation products exhibit chemotactic activities and activate pro-inflammatory cytokines, and induce mixed lesions called steatohepatitis. Therefore, an antioxidant strategy is needed to prevent the activation of pro-inflammatory cytokines.

Cite the following article in the introduction section.

https://doi.org/10.3390/app10186200.

Comment 3- The authors are advised to discuss liver regeneration and factors involved in the regulation of this process in a separate paragraph.

Comment 4- Mice lacking c-Jun in the liver have been reported to display impaired liver regeneration. It is advised to include a brief description of the molecular pathway involved in the liver regeneration regulated by c-Jun/AP-1 in the section “Transient transfection with HNF4α, c-Jun and silencing c-Jun expression”.

Comment 5- Include the name of the primer designing tool and its date of assessment in the methodology section.

Comment 6-Western blot is an important strategy used to determine the expression of proteins related to inflammation, oxidative stress, apoptosis, endoplasmic reticulum stress (ERS), and MAPK signaling. In the section SDS-PAGE and immunoblotting, briefly discuss the significance of immunoblotting in this study.

Comment 7- Did the authors determine the intracellular reactive oxygen content and oxidative stress-related markers?

Comment 8- The conclusions are too few. I think they should be expanded with future perspectives.

Comment 9- The manuscript should be strictly checked for grammar. Some sentences are not grammatically sound.

Comment 10- The authors have not provided the significance of this study. In which particular aspect, this study is different from the rest published so far.

The manuscript should be strictly checked for grammar. Some sentences are not grammatically sound.

Author Response

  • “Comment 1- Start the introduction section with a brief role of the liver in metabolic homeostasis.”

We now have included this information in the edited version of our manuscript and re-wrote the paragraph in the introduction:

“The liver plays a critical role for numerous physiological processes including e.g., synthesis of proteins, hormones and bile acids, catabolism of xenobiotics, metabolism of carbohydrates, cholesterol and lipids. Hepatocytes represent the liver cell type fulfilling the majority of these functions and thereby reflecting their important role in maintaining metabolic homeostasis [1]. Because of this indispensable role for the body, the liver holds a tremendous capacity for regeneration including anti-oxidative and anti-apoptotic mechanisms [2,3].”

  • “Comment 2- Oxidative stress plays an important role in liver cirrhosis and related pathologies. The lipid peroxidation products exhibit chemotactic activities and activate pro-inflammatory cytokines, and induce mixed lesions called steatohepatitis. Therefore, an antioxidant strategy is needed to prevent the activation of pro-inflammatory cytokines.”

We now have included this information in the edited version of our manuscript and re-wrote parts of the discussion by pointing to the subject: e.g.

“Oxidative stress and its resulting lipid peroxidation products may activate pro-inflammatory cytokines and are involved in fibrogenesis. Therefore, altering the anti-oxidative capacity may prevent the activation of pro-inflammatory cytokines [39].”

“Therefore, enhancing the expression of ALR in hepatocytes may increase the cellular anti-oxidative defense, lower activation of pro-inflammatory cytokines and might protect hepatocytes against apoptosis during NASH or cholestasis.”

  • “Comment 3- The authors are advised to discuss liver regeneration and factors involved in the regulation of this process in a separate paragraph.”

We are aware of this concern, but this issue was described in detail in several excellent reviews over the last years (N. Fausto, G.K. Michalopoulos and others). Because we do not focus on the process of liver regeneration, but on the molecular mechanisms of ALR regulation (transcription, translation), we suppose this concern is out of scope of the study.  

  • “Comment 4- Mice lacking c-Jun in the liver have been reported to display impaired liver regeneration. It is advised to include a brief description of the molecular pathway involved in the liver regeneration regulated by c-Jun/AP-1 in the section “Transient transfection with HNF4α, c-Jun and silencing c-Jun expression”.

We now have included this information in the edited version of our manuscript and re-wrote a paragraph of the discussion:

“. Interestingly, in contrast to results obtained from the referenced promoter study [36], silencing or enhancing c-Jun expression did not change ALR mRNA or protein expression as well as its IL-1ß-mediated reduction. Therefore, ALR expression as well as its reduction by IL-1ß is independent of c-Jun expression. Of note, blunted c-Jun expression reduces cell proliferation and c-Jun knock-out mice show impaired liver regeneration mainly based on down-regulation of tumor suppressors p53 and p21 and their transcriptional activities. [46,47] The findings of c-Jun-independent ALR expression in this study are not contradictory to these reports, unlike results from the ALR promoter study [36], and underline the importance of the performed protein expression analysis.”

  • “Comment 5- Include the name of the primer designing tool and its date of assessment in the methodology section.”

We now have included this information in the edited version and added references and the primer designing tool in the corresponding paragraph of section ‘material and method’.

  • “Comment 6-Western blot is an important strategy used to determine the expression of proteins related to inflammation, oxidative stress, apoptosis, endoplasmic reticulum stress (ERS), and MAPK signaling. In the section SDS-PAGE and immunoblotting, briefly discuss the significance of immunoblotting in this study.”

We are aware of this point, and fully agree that results and their conclusions based on protein expression are scientifically much more solid and evident compared to data obtained from mRNA or promoter activity analysis. Therefore, throughout the manuscript we emphasized the importance of protein expression analysis. e.g. “… and that c-Jun binds to an AP-1/AP-4 binding site (-375/-369) within ALR promoter repressing its activity in HepG2 cells based on a promotor study [36]. In this study we could show that IL-1ß induces c-Jun mRNA expression, c-Jun phosphorylation as well as nuclear localization in several hepatoma cells in contrast to AP-4. Interestingly, in contrast to results obtained from the referenced promoter study [36], silencing or enhancing c-Jun expression did not change ALR mRNA or protein expression as well as its IL-1ß-mediated reduction.”

  • “Comment 7- Did the authors determine the intracellular reactive oxygen content and oxidative stress-related markers?”

The focus of this study are the molecular mechanisms of ALR regulation and the link between inflammation (examples of cholestasis and NAFLD) and ALR regulation. By adding too many subjects, the clarity of the story might be lost and the findings in the manuscript may be diluted. Nevertheless, the concern raised by the reviewer is of importance and will be the subject of future studies.

  • “Comment 8- The conclusions are too few. I think they should be expanded with future perspectives.”
  • “Comment 10- The authors have not provided the significance of this study. In which particular aspect, this study is different from the rest published so far.”

We now have included this information of both of the comments in the edited version of our manuscript and re-wrote the last paragraph of the discussion:

“While most researchers focused on the role of ALR in disease development by performing loss and gain of function models, only less is known about the molecular mechanisms of ALR regulation. Here we present evidence, that IL-1ß reduces ALR mRNA and protein expression by diminished HNF4a expression and decreased Egr-1 binding to ALR promoter independent of c-Jun activation. ALR was reported to have anti‑apoptotic and anti‑oxidative properties resulting in less oxidative stress and presumably less induction of pro-inflammatory cytokines. Even though ALR expression is induced by oxidative stress, excessive levels of IL-1ß might impede proper ALR expression for cell protection and regeneration. Low ALR expression levels are reported in liver tissue from patients with NASH or cholestasis and the observed high IL-1ß levels in these patients may be causative. Further studies investigating the impact of IL-1ß on ALR’s hepatoprotective capacity, expression of ALR under inflammatory conditions and correlations of IL-1ß and ALR levels in patients would be of interest.”

  • “Comment 9- The manuscript should be strictly checked for grammar. Some sentences are not grammatically sound.”

The manuscript was read by a native English speaker and corrections were done.